# Algorithmic Guarantees for Inverse Imaging with Untrained Network Priors

**Gauri Jagatap**
New York University
gauri.jagatap@nyu.edu

**Chinmay Hegde**
New York University
chinmay.h@nyu.edu

## Abstract

Deep neural networks as image priors have been recently introduced for problems such as denoising, super-resolution and inpainting with promising performance gains over hand-crafted image priors such as sparsity. Unlike *learned* generative priors they do not require any training over large datasets. However, few theoretical guarantees exist in the scope of using untrained network priors for inverse imaging problems. We explore new applications and theory for untrained neural network priors. Specifically, we consider the problem of solving linear inverse problems, such as compressive sensing, as well as non-linear problems, such as compressive phase retrieval. We model images to lie in the range of an untrained deep generative network with a fixed seed. We further present a projected gradient descent scheme that can be used for both compressive sensing and phase retrieval and provide rigorous theoretical guarantees for its convergence. We also show both theoretically as well as empirically that with deep neural network priors, one can achieve better compression rates for the same image quality as compared to when hand crafted priors are used.

## 1 Introduction

### 1.1 Motivation

Deep neural networks have led to unprecedented success in solving several problems, specifically in the domain of inverse imaging. Image denoising [1], super-resolution [2], inpainting and compressed sensing [3], and phase retrieval [4] are among the many imaging applications that have benefited from the usage of deep convolutional networks (CNNs) trained with thousands of images.

Apart from supervised learning, deep CNN models have also been used in unsupervised setups, such as Generative Adversarial Networks (GANs). Here, image priors based on a generative model [5] are learned from training data. In this context, neural networks emulate the probability distribution of the data inputs. GANs have been used to model signal prior by learning the distribution of training data. Such learned priors have replaced hand-crafted priors with high success rates [3, 6, 7, 8]. However, the main challenge with these approaches is the requirement of massive amounts of training data. For instance, super-resolution CNN [2] uses ImageNet which contains millions of images. Moreover, convergence guarantees for training such networks are limited [7].

In contrast, there has been recent interest in using *untrained* neural networks as an image prior. Deep Image Prior [9] and variants such as Deep Decoder [10] are capable of solving linear inverse imaging problems with no training data whatsover, while merely imposing an auto-encoder [9] and decoder [10] architecture as a structural prior. For denoising, inpainting and super-resolution, deep image priors have shown superior reconstruction performance as compared to conventional methodologies such as basis pursuit denoising (BPDN) [11], BM3D [12] as well as convolutional sparse coding [13]. Similar emperical results have been claimed very recently in the context of time-series data

for audio applications [14, 15]. The theme in all of these approaches is the same: to design a prior that exploits *local* image correlation, instead of global statistics, and find a good low-dimensional *neural* representation of natural images. However, most of these works have very limited [16, 10] or no theoretical guarantees.

Neural networks priors for compressive imaging has only recently been explored. In the context of compressive sensing (CS), [17] uses Deep Image Prior along with *learned regularization* for reconstructing images from compressive measurements [18]. However, the model described still relies on training data for learning appropriate regularization parameters. For the problem of compressive sensing, priors such as sparsity [19] and structured sparsity [20] have been traditionally used.

Phase retrieval is another inverse imaging problem in several Fourier imaging applications, which involves reconstructing images from magnitude-only measurements. Compressive phase retrieval (CPR) models use sparse priors for reducing sample requirements; however, standard techniques from recent literature [21] suggest a quadratic dependence of number of measurements on the sparsity level for recovering sparse images from magnitude-only Gaussian measurements and the design of a smart initialization scheme [22, 21]. If a prior is learned via a GAN [7], [23], then this requirement can be brought down; however one requires sufficient training data, which can be prohibitively expensive to obtain in domains such as medical or astronomical imaging.

## 1.2 Our contributions

In this paper, we explore, in depth, the use of untrained deep neural networks as an image prior for inverting images from under-sampled linear and non-linear measurements. Specifically, we assume that the image, $x^{*d \times 1}$ has $d$ pixels. We further assume that the image $x^*$ belongs to the range spanned by the weights of a deep *under-parameterized* untrained neural network $G(\mathbf{w}; z)$, which we denote by $\mathcal{S}$, where $\mathbf{w}$ is a set of the weights of the deep network and $z$ is the latent code. The compressive measurements are stored in vector $y = f(x^*)$, where $f$ embeds either compressive linear (defined by operator $A(\cdot)$) or compressive magnitude-only (defined by operator $|A(\cdot)|$) measurements. The task is to reconstruct image $\hat{x}$ which corresponds to small measurement error $\min_{x \in \mathcal{S}} \|f(x) - y\|_2^2$. With this setup, we establish theoretical guarantees for successful image reconstruction from both measurement schemes under untrained network priors.

Our specific contributions are as follows:

- We first present a new variant of the Restricted Isometry Property (RIP) [18] via a covering number argument for the range of images $\mathcal{S}$ spanned by a deep untrained neural network. We use this result to guarantee unique image reconstruction for two different compressive imaging schemes.

- We propose a projected gradient descent (PGD) algorithm for solving the problem of compressive sensing with a deep untrained network prior. To our knowledge this is the first paper to use deep neural network priors for compressive sensing [1], which relies on no training data[2]. We analyze the conditions under which PGD provably converges and report the sample complexity requirements corresponding to it. We also show superior performance of this framework via empirical results.

- We are the first to use deep network priors in the context of phase retrieval. We introduce a novel formulation, to solve compressive phase retrieval with fewer measurements as compared to state-of-art. We further provide preliminary guarantees for the convergence of a projected gradient descent scheme to solve the problem of compressive phase retrieval. We empirically show significant improvements in image reconstruction quality as compared to prior works.

We note that our sample complexity results rely on the number of parameters of the assumed deep network prior. Therefore, to get meaningful bounds, our network priors are *under-parameterized*, in that the total number of unknown parameters of the deep network is smaller than the dimension of the image. To ensure this, we build upon the formulation of the deep decoder [10], which is a special network architecture resembling the decoder of an autoencoder (or generator of a GAN). The requirement of under-parameterization of deep network priors is natural; the goal is to design priors that *concisely* represent natural images. Moreover, this also ensures that the network does not fit noise [10]. Due to these merits, we use select the deep decoder architecture for all analyses in this paper.

## 1.3 Prior work

Sparsifying transforms have long been used to constrain the solutions of inverse imaging problems in the context of denoising or inpainting. Conventional approaches to solve these problems include Basis Pursuit Denoising (BPDN) or Lasso [11], TVAL3 [25], which rely on using $\ell_0$, $\ell_1$ and total variation (TV) regularizations on the image to be recovered. Sparsity based priors are highly effective and dataset independent, however it heavily relies on choosing a good sparsifying basis [26].

Instead of hand-picking the sparsifying transform, in dictionary learning one learns both the sparsifying transform and the sparse code [27]. The dictionary captures global statistics of a given dataset [3]. Multi-layer convolutional sparse coding [16] is an extension of sparse coding which models a given dataset in the form of a product of several linear dictionaries, all of which are convolutional in nature and this problem is challenging.

Generative adversarial networks (GAN) [5] have been used to generate photo-realistic images in an unsupervised fashion. The generator consists of stacked convolutions and maps random low-dimensional noise vectors to full sized images. GAN priors have been successfully used for inverse imaging problems [6, 7, 28, 29, 8]. The shortcomings of this approach are two-fold: test images are strictly restricted to the range of a trained generator, and the requirement of sufficient training data.

Sparse signal recovery from linear compressive measurements [18] as well as magnitude-only compressive measurements [21] has been extensively studied, with several algorithmic approaches [19, 21]. In all of these approaches, modeling the low-dimensional embedding is challenging and may not be captured correctly using simple hand-crafted priors such as structured sparsity [20]. Since it is hard to estimate these hyper-parameters accurately, the number of samples required to reconstruct the image is often much higher than information theoretic limits [30, 6].

The problem of compressive phase retrieval specifically, is even more challenging because it is non-convex. Several papers in recent literature [31, 32, 21] rely on the design of a spectral initialization scheme which ensures that one can subsequently optimize over a convex ball of the problem. However this initialization requirement results in high sample requirements and is a bottleneck in achieving information theoretically optimal sample complexity.

Deep image prior [9] (DIP) uses primarily an encoder-decoder as a *prior* on the image, alongside an early stopping condition, for inverse imaging problems such as denoising, super-resolution and inpainting. Deep decoder [10] (DD) improves upon DIP, providing a much simpler, *underparameterized* architecture, to learn a low-dimensional manifold (latent code) and a decoding operation from this latent code to the full image. Because it is under parameterized, deep decoder does not fit noise, and therefore does not require early stopping.

Deep network priors in the context of compressive imaging have only recently been explored [17], and only in the context of compressive sensing. In contrast with [17] which extends the idea of a Deep Image Prior to incorporate learned regularizations, in this paper we focus more on theoretical aspects of the problem and also explore applications in compressive phase retrieval. To our knowledge the application of deep network priors to compressive phase retrieval is novel.

## 2 Notation

Throughout the paper, lower case letters denote vectors, such as $v$ and upper case letters for matrices, such as $M$. A set of variables subscripted with different indices is represented with bold-faced shorthand of the following form: $\mathbf{w} := \{W_1, W_2, \ldots W_L\}$. The neural network consists of $L$ layers, each layer denoted as $W_l$, with $l \in \{1, \ldots L\}$ and are $1 \times 1$ convolutional. Up-sampling operators are denoted by $U_l$. Vectorization of a matrix is written as $\mathrm{vec}(\cdot)$. The activation function considered is Rectified Linear Unit (ReLU), denoted as $\sigma(\cdot)$. Hadamard or element-wise product is denoted by $\circ$. Element-wise absolute valued vector is denoted by $|v|$. Unless mentioned otherwise, $\|v\|$ denotes vector $\ell_2$-norm and $\|M\|$ denotes spectral norm $\|M\|_2$.

# 3 Problem setup

## 3.1 Deep neural network priors

In this paper we discuss the problem of inverting a mapping $x \rightarrow y$ of the form:

$$y = f(x)$$

where $x = \text{vec}(X)^{dk}$ is a $d$-dimensional signal $X^{d \times k}$ (vectorized image), with $k$ channels and $f : x \rightarrow y \in \mathbb{R}^n$ captures a compressive measurement procedure, such as a linear operator $A(\cdot)$ or magnitude only measurements $|A(\cdot)|$ and $n < dk$. We elaborate further on the exact structure of $f$ in the next subsection (Section 3.2). The task of reconstructing image $x$ from measurements $y$ can be formulated as an optimization problem of the form:

$$\min_{x \in S} \|y - f(x)\|^2 \tag{1}$$

where we have chosen the $\ell_2$-squared loss function and where $\mathcal{S}$ captures the prior on the image.

If the image $x$ can be represented as the action of a deep generative network $G(\mathbf{w}; z)$ with weights $\mathbf{w}$ on some latent code $z$, such that $x = G(\mathbf{w}; z)$, then the set $\mathcal{S}$ captures the characteristics of $G(\mathbf{w}; z)$. The latent code $z := \text{vec}(Z_1)$ with $Z_1 \in \mathbb{R}^{d_1 \times k_1}$ is a low-dimensional embedding with dimension $d_1 k_1 \ll dk$ and its elements are generated from uniform random distribution.

When the network $G(\cdot)$ and its weights $\mathbf{w} := \{W_1, \dots W_L\}$ are *known* (from pre-training a generative network over large datasets) and fixed, the task is to obtain an estimate $\hat{x} = G(\mathbf{w}; \hat{z})$, which indirectly translates to finding the optimal latent space encoding $\hat{z}$. This problem has been studied in [6, 7] in the form of using learned GAN priors for inverse imaging.

In this paper however, the weights of the generator $\mathbf{w}$ are *not pre-trained*; rather, the task is to estimate image $\hat{x} = G(\hat{\mathbf{w}}; z) \approx G(\mathbf{w}^*; z) = x^*$ and corresponding weights $\hat{\mathbf{w}}$, for a *fixed* seed $z$, where $x^*$ is assumed to be the true image and the true weights $\mathbf{w}^*$ (possibly non-unique) satisfy $\mathbf{w}^* = \min_{\mathbf{w}} \|x^* - G(\mathbf{w}; z)\|_2^2$. Note that the optimization in Eq. 1 is equivalent to substituting the surjective mapping $G : \mathbf{w} \rightarrow x$, and optimizing over $\mathbf{w}$,

$$\min_{\mathbf{w}} \|y - f(G(\mathbf{w}; z))\|^2, \tag{2}$$

and estimate weights $\hat{\mathbf{w}}$ and corresponding image $\hat{x}$.

Specifically, the untrained network $G(\mathbf{w}; z)$ takes the form of an expansive neural network; a decoder architecture similar to the one in [10] [4]. The neural network is composed of $L$ weight layers $W_l$, indexed by $l \in \{1, \dots, L\}$ and are $1 \times 1$ convolutions, upsampling operators $U_l$ for $l \in \{1, \dots L-1\}$ and ReLU activation $\sigma(\cdot)$ and is expressed as follows

$$x = G(\mathbf{w}; z) = U_{L-1} \sigma(Z_{L-1} W_{L-1}) W_L = Z_L W_L, \tag{3}$$

where $\sigma(\cdot)$ represents the action of ReLU operation, $Z_i^{d_i \times k_i} = U_{i-1} \sigma(Z_{i-1} W_{i-1})$, for $i = 2, \dots L$, $z = \text{vec}(Z_1)$, $d_L = d$ and $W_L \in \mathbb{R}^{k_L \times k}$.

To capture the range of images spanned by the deep neural network architecture described above, we formally introduce the main assumption in our paper through Definition 1. Without loss in generality, we set $k = 1$ for the rest of this paper, while noting that the techniques carry over to general $k$.

**Definition 1.** *A given image $x \in \mathbb{R}^d$ is said to obey an untrained neural network prior if it belongs to a set $\mathcal{S}$ defined as:*

$$\mathcal{S} := \{x | x = G(\mathbf{w}; z)\}$$

*where $z$ is a (randomly chosen, fixed) latent code vector and $G(\mathbf{w}; z)$ has the form in Eq. 3.*

## 3.2 Observation models and assumptions

We now discuss the compressive measurement setup in more detail. Compressive measurement schemes were developed in [18] for efficient imaging and storage of images and work only as long as certain structural assumptions on the signal (or image) are met. The optimization problem in Eq.1 is

non-convex in general, partly dictated by the non-convexity of set $\mathcal{S}$. Moreover, in the case of phase retrieval, the loss function is itself non-convex. Therefore unique signal recovery for either problems is not guaranteed without making specific assumptions on the measurement setup.

In this paper, we assume that the measurement operation can be represented by the action of a Gaussian matrix $A$ which is rank-deficient ($n < d$). The entries of this matrix are such that $A_{ij} \sim \mathcal{N}(0, 1/n)$. Linear compressive measurements take the form $y = Ax$ and magnitude-only measurements take the form $y = |Ax|$. We formally discuss the two different imaging schemes in the next two sections. We also present algorithms and theoretical guarantees for their convergence. For both algorithms, we require that a special $(\mathcal{S}, \gamma, \beta)$-RIP holds for measurement matrix $A$, which is defined below.

**Definition 2.** $(\mathcal{S}, \gamma, \beta)$-*RIP: Set-Restricted Isometry Property with parameters* $\gamma, \beta$:

*For parameters* $\gamma, \beta > 0$, *a matrix* $A \in \mathbb{R}^{n \times d}$ *satisfies* $(\mathcal{S}, \gamma, \beta)$-*RIP, if for all* $x \in \mathcal{S}$,

$$\gamma \|x\|^2 \leq \|Ax\|^2 \leq \beta \|x\|^2.$$

*We refer to the left (lower) inequality as* $(\mathcal{S}, \gamma)$-*RIP and right (upper) inequality as* $(\mathcal{S}, \beta)$-*RIP.*

The $(\mathcal{S}, 1 - \alpha, 1 + \alpha)$ RIP is achieved by Gaussian matrix $A$ under certain assumptions, which we state and prove via Lemma 1 as follows.

**Lemma 1.** *If an image* $x \in \mathbb{R}^d$ *has a decoder prior (captured in set* $\mathcal{S}$*), where the decoder consists of weights* $\mathbf{w}$ *and piece-wise linear activation (ReLU), a random Gaussian matrix* $A \in \mathbb{R}^{n \times d}$ *with elements from* $\mathcal{N}(0, 1/n)$*, satisfies* $(\mathcal{S}, 1 - \alpha, 1 + \alpha)$-*RIP, with probability* $1 - e^{-c\alpha^2 n}$*, as long as* $n = O\left( \frac{k_1}{\alpha^2} \sum_{l=2}^{L} k_l \log d \right)$*, for small constant* $c$ *and* $0 < \alpha < 1$.

***Proof sketch:*** We use a union of sub-spaces model, similar to that developed in [6] which was developed for GAN priors, to capture the range of a deep untrained network.

Our method uses a *linearization principle*. If the output sign of any ReLU activation $\sigma(\cdot)$ on its inputs were known *a priori*, then the mapping $x = G(\mathbf{w}; z)$ becomes a product of linear weight matrices and linear upsampling operators acting on the latent code $z$. The bulk of the proof relies on constructing a counting argument for the number of such linearized networks; call that number $N$. For a fixed linear subspace, the image $x$ has a representation of the form $x = UZw$, where $U$ absorbs all upsampling operations, $Z$ is latent code which is fixed and known and $w$ is the direct product of all weight matrices with $w \in \mathbb{R}^{k_1}$. An oblivious subspace embedding (OSE) of $x$ takes the form

$$(1 - \alpha)\|x\|^2 \leq \|Ax\|^2 \leq (1 + \alpha)\|x\|^2,$$

where $A$ is a Gaussian matrix, and holds for all $k_1$-dimensional vectors $w$, with high probability as long as $n = O(k_1/\alpha^2)$. We further require to take a union bound over all possible such linearized networks, which is given by $N$. The sample complexity corresponding to this bound is then computed to complete the set-restricted RIP result. The complete proof can be found in Appendix D and a discussion on the sample complexity is presented in Appendix B.

## 4 Linear compressive sensing with deep network prior

We now analyze linear compressed Gaussian measurements of a vectorized image $x$, with a deep network prior. The reconstruction problem assumes the following form:

$$\min_{x} \quad \|y - Ax\|^2 \quad \text{s.t.} \quad x = G(\mathbf{w}; z), \tag{4}$$

where $A \in \mathbb{R}^{n \times d}$ is Gaussian matrix with $n < d$, unknown weight matrices $\mathbf{w}$ and latent code $z$ which is fixed. We solve this problem via Algorithm 1, Network Projected Gradient Descent (Net-PGD) for compressed sensing recovery.

Specifically, we break down the minimization into two parts; we first solve an unconstrained loss minimization of the objective function in Eq. 4 by implementing one step of gradient descent in Step 3 of Algorithm 1. The update $v^t$ typically does not adhere to the deep network prior constraint $v^t \notin \mathcal{S}$. To ensure that this happens, we solve a projection step in Line 4 of Algorithm 1, which happens to be the same as fitting a deep network prior to a noisy image. We iterate through this procedure in an alternating fashion until the estimates $x^t$ converge to $x^*$ within error factor $\epsilon$.

We further establish convergence guarantees for Algorithm 1 in Theorem 1.

---

**Algorithm 1** Net-PGD for compressed sensing recovery.

---

1: **Input:** $y, A, z = \text{vec}(Z_1), \eta, T = \log \frac{1}{\epsilon}$
2: **for** $t = 1, \cdots, T$ **do**
3:    $v^t \leftarrow x^t - \eta A^\top (Ax^t - y)$          {gradient step for least squares}
4:    $\mathbf{w}^t \leftarrow \arg\min_{\mathbf{w}} \|v^t - G(\mathbf{w}; z)\|$      {projection to range of deep network}
5:    $x^{t+1} \leftarrow G(\mathbf{w}^t; z)$
6: **end for**
7: **Output** $\hat{x} \leftarrow x^T$.

---

**Theorem 1.** *Suppose the sampling matrix $A^{n \times d}$ satisfies $(\mathcal{S}, 1 - \alpha, 1 + \alpha)$-RIP with high probability then, Algorithm 1, with $\eta$ small enough, produces $\hat{x}$ such that $\|\hat{x} - x^*\| \leq \epsilon$ and requires $T \propto \log \frac{1}{\epsilon}$ iterations.*

***Proof sketch:*** The proof of this theorem predominantly relies on our new set-restricted RIP result and uses standard techniques from compressed sensing theory. Indicating the loss function in Eq. 4 as $L(x^t) = \|y - Ax^t\|^2$, we aim to establish a contraction of the form $L(x^{t+1}) < \nu L(x^t)$, with $\nu < 1$. To achieve this, we combine the projection criterion in Step 4 of Algorithm 1, which strictly implies that

$$\|x^{t+1} - v^t\| \leq \|x^* - v^t\|$$

and $v^t = x^t - \eta A^\top (Ax^t - y)$ from Step 3 of Algorithm 1, where $\eta$ is chosen appropriately. Therefore,

$$\|x^{t+1} - x^t + \eta A^\top A (x^t - x^*)\|^2 \leq \|x^* - x^t + \eta A^\top A (x^t - x^*)\|^2.$$

Furthermore, we utilize $(\mathcal{S}, 1 - \alpha, 1 + \alpha)$-RIP and its Corollary 1 (refer Appendix D) which apply to $x^*, x^t, x^{t+1} \in \mathcal{S}$, to show that

$$L(x^{t+1}) \leq \nu L(x^t)$$

and subsequently the error contraction $\|x^{t+1} - x^*\| \leq \nu_o \|x^t - x^*\|$, with $\nu, \nu_o < 1$ to guarantee linear convergence of Net-PGD for compressed sensing recovery. This convergence result implies that Net-PGD requires $T \propto \log 1/\epsilon$ iterations to produce $\hat{x}$ within $\epsilon$-accuracy of $x^*$. The complete proof of Theorem 1 can be found in Appendix D. In Appendix A we provide some exposition on the projection step (line 4 of Algorithm 1).

## 5   Compressive phase retrieval under deep image prior

In compressive phase retrieval, one wants to reconstruct a signal $x \approx x^* \in \mathcal{S}$ from measurements of the form $y = |Ax^*|$ and therefore the objective is to minimize the following

$$\min_x \quad \|y - |Ax|\|^2 \quad \text{s.t.} \quad x = G(\mathbf{w}; z), \tag{5}$$

where $n < d$ and $A$ is Gaussian, $z$ is a fixed seed and weights $\mathbf{w}$ need to be estimated. We propose a Network Projected Gradient Descent (Net-PGD) for compressive phase retrieval to solve this problem, which is presented in Algorithm 2.

Algorithm 2 broadly consists of two parts. For the first part, in Line 3 we estimate the phase of the current estimate and in Line 4 we use this to compute the Wirtinger gradient [31] and execute one step for solving an unconstrained phase retrieval problem with gradient descent. The second part of the algorithm is (Line 5), estimating the weights of the deep network prior with noisy input $v^t$. This is the projection step and ensures that the output $\mathbf{w}^t$ and subsequently the image estimate $x^t = G(\mathbf{w}^t; z)$ lies in the range of the decoder $G(\cdot)$ outlined by set $\mathcal{S}$.

We highlight that the problem in Eq. 5 is significantly more challenging than the one in Eq. 4. The difficulty hinges on estimating the missing phase information accurately. For a real-valued vectors, there are $2^n$ different phase vectors $p = \text{sign}(Ax)$ for a fixed choice of $x$, which satisfy $y = |Ax|$, moreover the entries of $p$ are restricted to $\{1, -1\}$. Hence, phase estimation is a non-convex problem. Therefore, with Algorithm 2 the problem in Eq.5 can only be solved to convergence locally; an initialization scheme is required to establish global convergence guarantees. We highlight the guarantees of Algorithm 2 in Theorem 2.

---

**Algorithm 2** Net-PGD for compressive phase retrieval.

---

1: **Input:** $A, z = \text{vec}(Z_1), \eta, T = \log \frac{1}{\epsilon}, x^0$ s.t. $\|x^0 - x^*\| \leq \delta_i \|x^*\|$.
2: **for** $t = 1, \cdots, T$ **do**
3:    $p^t \leftarrow \text{sign}(Ax^t)$                                    {phase estimation}
4:    $v^t \leftarrow x^t - \eta A^\top (Ax^t - y \circ p^t)$              {gradient step for phase retrieval}
5:    $\mathbf{w}^t \leftarrow \arg\min_{\mathbf{w}} \|v^t - G(\mathbf{w}; z)\|$    {projection to range of deep network}
6:    $x^{t+1} \leftarrow G(\mathbf{w}^t; z)$
7: **end for**
8: **Output** $\hat{x} \leftarrow x^T$.

---

**Theorem 2.** *Suppose the sampling matrix $A^{n \times d}$ with Gaussian entries satisfies $(\mathcal{S}, 1-\alpha, 1+\alpha)$-RIP with high probability, Algorithm 2 solves Eq. 5 with $\eta$ small enough, such that $\|\hat{x} - x^*\| \leq \epsilon$, as long as the weights are initialized appropriately and the number of measurements is $n = O\left(k_1 \sum_{l=2}^{L} k_l \log d\right)$.*

***Proof sketch:*** The proof for Theorem 2 relies on two important results; $(\mathcal{S}, 1-\alpha, 1+\alpha)$-RIP and Lemma 2 which establishes a bound on the phase estimation error. Formally, the update in Step 4 of Algorithm 2 can be re-written as

$$v^{t+1} = x^t - \eta A^\top \left(Ax^t - Ax^* \circ \text{sign}(Ax^*) \circ \text{sign}(Ax^t)\right) = x^t - \eta A^\top \left(Ax^t - Ax^*\right) - \eta \varepsilon_p^t$$

where $\varepsilon_p^t := A^\top Ax^* \circ (1 - \text{sign}(Ax^*) \circ \text{sign}(Ax^t))$ is *phase estimation* error.

If $\text{sign}(Ax^*) \approx \text{sign}(Ax^t)$, then the above resembles the gradient step from the linear compressive sensing formulation. Thus, if $x^0$ is initialized well, the error due to phase mis-match $\varepsilon_p^t$ can be bounded, and subsequently, a convergence result can be formulated.

Next, Step 5 of Algorithm 2 learns weights $\mathbf{w}^t$ that produce $x^t = G(\mathbf{w}^t; z)$, such that

$$\|x^{t+1} - v^t\| \leq \|x^t - v^t\|$$

for $t = \{1, 2, \ldots T\}$. Then, the above projection rule yields:

$$\|x^{t+1} - v^{t+1} + v^{t+1} - x^*\| \leq \|x^{t+1} - v^{t+1}\| + \|x^* - v^{t+1}\| \leq 2\|x^* - v^{t+1}\|,$$

Using the update rule from Eq. 12 and plugging in for $v^{t+1}$:

$$\frac{1}{2}\|x^{t+1} - x^*\| \leq \|(1 - \eta A^\top A)h^t\| + \|\varepsilon_p^t\|$$

where $\eta$ is chosen appropriately. The rest of the proof relies on bounding the first term via matrix norm inequalities using Corollary 2 (in Appendix D) of $(\mathcal{S}, 1-\alpha, 1+\alpha)$-RIP as $\|(1-\eta A^\top A)h^t\| \leq \rho_o \|h^t\|$ and the second term is bounded via Lemma 2 as $\|\varepsilon_p^t\| \leq \delta_o \|x^t - x^*\|$ as long as $\|x^0 - x^*\| \leq \delta_i \|x^*\|$. Hence we obtain a convergence criterion of the form

$$\|x^{t+1} - x^*\| \leq 2(\rho_o + \eta \delta_o)\|x^t - x^*\| := \rho \|x^t - x^*\|.$$

where $\rho < 1$. Note that this proof relies on a bound on the phase error $\|\varepsilon_p^t\|$ which is established via Lemma 2. The complete proof for Theorem 2 can be found in Appendix D. In Appendix A we provide some exposition on the projection step (line 5 of Algorithm 2). In our experiments (Section 6) we note that a uniform random initialization of the weights $\mathbf{w}^0$ (which is common in training neural networks), to yield $x^0 = G(\mathbf{w}^0; z)$ is sufficient for Net-PGD to succeed for compressive phase retrieval. In Appendix C we show experimental evidence to support this claim.

## 6  Experimental results

*Dataset:* We use images from the MNIST database and CelebA database to test our algorithms and reconstruct 6 grayscale (MNIST, $28 \times 28$ pixels ($d = 784$)) and 5 RGB (CelebA) images. The CelebA dataset images are center cropped to size $64 \times 64 \times 3$ ($d = 12288$). The pixel values of all images are scaled to lie between 0 and 1.

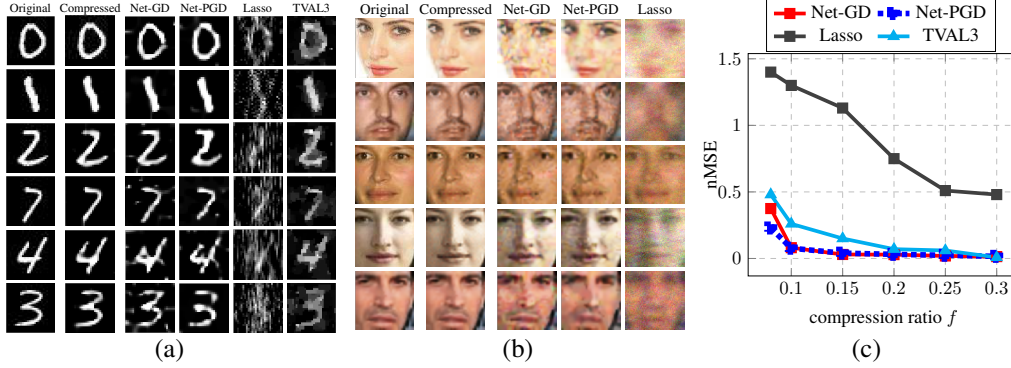

Figure 1: (CS) Reconstructed images from linear measurements (at compression rate $n/d = 0.1$) with (a) $n = 78$ measurements for examples from MNIST, (b) $n = 1228$ measurements for examples from CelebA, and (c) nMSE at different compression rates $f = n/d$ for MNIST.

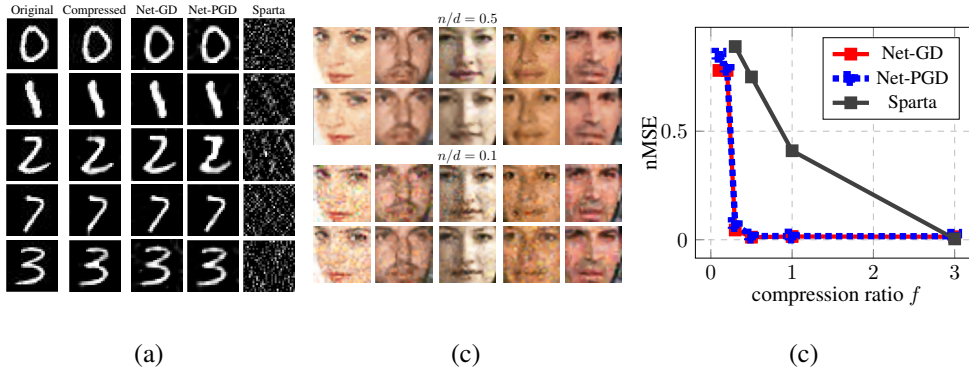

Figure 2: (CPR) Reconstructed images from magnitude-only measurements (a) at compression rate of $n/d = 0.3$ for MNIST, (b) at compression rates of $n/d = 0.1, 0.5$ for CelebA with (row 1,3) Net-GD and (row 2,4) Net-PGD, (c) nMSE at different compression rates $f = n/d$ for MNIST.

*Deep network architecture:* We first optimize the deep network architecture which fit our example images such that $x^* \approx G(\mathbf{w}^*; z)$ (referred as "compressed" image). For MNIST images, the architecture was fixed to a 2 layer configuration $k_1 = 15, k_2 = 15, k_3 = 10$, and for CelebA images, a 3 layer configuration with $k_1 = 120, k_2 = 15, k_3 = 15, k_4 = 10$. Both architectures use bilinear upsampling operations. Further details on this setup can be found in Appendix C.

*Measurement setup:* We use a Gaussian measurement matrix of size $n \times d$ with $n$ varied such that (i) $n/d = 0.08, 0.1, 0.15, 0.2, 0.25, 0.3$ for compressive sensing and (ii) $n/d = 0.1, 0.2, 0.3, 0.5, 1, 3$ for compressive phase retrieval. The elements of $A$ are picked such that $A_{i,j} \sim \mathcal{N}(0, 1/n)$ and we report averaged reconstruction error values over 10 different instantiations of $A$ for a fixed image (image of digit '0' from MNIST), network configuration and compression ratio $n/d$.

## 6.1 Compressive sensing

*Algorithms and baselines:* We implement 4 schemes based on *untrained* priors for solving CS, (i) gradient descent with deep network prior which solves Eq.2 (we call this Net-GD), similar to [17] but without learned regularization (ii) Net-PGD, (iii) Lasso ($\ell_1$ regularization) with sparse prior in DCT basis and finally (iv) TVAL3 [25] (Total Variation regularization). The TVAL3 code only works for grayscale images, therefore we do not use it for CelebA examples. The reconstructions are shown in Figure 1 for images from (a) MNIST and (b) CelebA datasets. The implementation details can be found in Appendix C.

*Performance metrics:* We compare reconstruction quality using normalized Mean-Squared Error (nMSE), which is calculated as $\|\hat{x} - x^*\|^2/\|x^*\|^2$. We plot the variation of the nMSE with different compression rates $f = n/d$ for all the algorithms tested averaged over all trials for MNIST in Figure 1 (c). We note that both Net-GD and Net-PGD produce superior reconstructions as compared to state of art. Running time performance is reported in Appendix C.

## 6.2   Compressive phase retrieval

*Algorithms and baselines:* We implement 3 schemes based on *untrained* priors for solving CPR , (i) Net-GD (ii) Net-PGD and finally (iii) Sparse Truncated Amplitude Flow (Sparta) [22], with sparse prior in DCT basis for both datasets. The reconstructions are shown in Figure 2 for (a) MNIST and (b) CelebA datasets. We plot nMSE at varying compression rates for all algorithms averaged over all trials for MNIST in Figure 2(c) and note that both Net-GD and Net-PGD outperform Sparta. Running term performance as well as goodness of random initialization scheme are discussed in Appendix C.

# 7   Acknowledgments

This work was supported in part by NSF grants CAREER CCF-2005804, CCF-1815101, and a faculty fellowship from the Black and Veatch Foundation.

## Footnotes

[1]We note recent concurrent work in [24] which explores a similar approach for compressive sensing; however our paper focuses theoretical guarantees rooted in an algorithmic procedure.

[2][17] requires training data for learning a regularization function.

[3] Local structural information from a single image can also be used to learn dictionaries, by constructing several overlapping crops or patches of a single image.

[4] Alternatively, one may assume the architecture of the generator of a DCGAN [33, 17].

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
