[Supplementary Material]

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

## A   Projection to deep network prior

The projection steps in both Algorithms 1 and 2 represent the problem of fitting an image to an untrained neural network representation. This is the original setting for denoising and compression applications in [9] and [10]. The algorithmic approach to solving this problem is via standard solvers such as gradient descent (GD) or Adam. The problem takes the form:

$$\min_{\mathbf{w}} \mathcal{L}(\mathbf{w}; z, v) := \min_{\mathbf{w}} \|v - G(\mathbf{w}; z)\|^2, \tag{6}$$

where $v$ is typically a noisy variant of the original image $x^*$. The problem in Eq.6 is non-convex due to the structure of $G(\mathbf{w}; z)$. Convergence guarantees for deep neural network formulations of this form that exist are highly restrictive [34, 35]. There exist several papers in recent literature which allude to (linear) convergence of gradient descent for solving the two-layer neural networks; however all of the results rely on moderate or extreme overparameterization of the neural network. Therefore, these results do not apply to our paper and deriving convergence guarantees for the denoising problem in 6 is an interesting direction for future work.

## B   Discussion on sample complexity

In compressive imaging literature, for $s$-sparse signals of dimension $d$, the sample complexity for compressive sensing is $n = O(s \log d)$ and compressive phase retrieval is $n = O(s^2 \log d)$, when Gaussian measurements are considered. If structural constraints are imposed on the sparsity of images, such as block sparsity, the sample requirements can be brought down to $n = O(s/b \log d)$ and $n = O(s^2/b \log d)$ for CS and CPR respectively, where $b$ is the block length of each sparse block [21]. However these gains come at the cost of designing the signal priors carefully.

In contrast, the sample requirements with deep network priors, as we show in this paper is $n = O(k_1 \sum_{l=2}^{L} k_l \log d)$. In both datasets that we tested, relatively shallow architectures were sufficient. Therefore the effective sample complexity is of the order of $k_1$, which is typically much smaller than the dimension $d$. We have empirically demonstrated in Section 6 that the sample requirement with deep network priors is significantly lower than that for the sparse prior setting. Moreover, the design of the prior is fairly straightforward, and applies for a wide class of images.

## C   Additional experiments

In this section we present some additional details for the experimental setup in Section 6. We also present some additional experiments to reinforce the merits of Net-PGD.

All codes were run on a Nvidia GeForce GPU with 8GB RAM.

*Deep network architecture:* For both MNIST and CelebA images, several architectures were tried out to pick out the best under-parameterized network which gave low representation error. We found that for the example images from MNIST, a decoder architecture, as described in Eq. 3 with 2 layers, and channel configurations $k_1 = 15, k_2 = 15, k_3 = 10$ and bilinear upsampling operators each with upsampling factor of 2, $U_l^{\uparrow 2}, l = \{1, 2, 3\}$ was sufficient to represent most images. The outputs after each ReLU operation are normalized, by calling for batch normalization subroutine in Pytorch. Finally a sigmoid activation is added to the output of the deep network, which smoothens the output; however this is not mandatory for the deep network configuration to work. For CelebA images, we fixed the configuration to a 3 layer network with setup $k_1 = 120, k_2 = 15, k_3 = 15, k_4 = 10$. Note that both of these architectures are *underparameterized*, unlike the configurations in [9]. The random seed $Z_1$ is fixed and picked from uniform random distribution [5]. We plot the "compressed" representations of each image, $G(\mathbf{w}; z)$ in all Figures for reference.

### C.1   Compressed sensing recovery

*Implementation details:* For CS recovery with deep network priors, both Net-GD and Net-PGD were implemented using the PyTorch framework with Python 3 and using GPU support. For Net-GD, SGD

(alternatively, Adam) optimizer is used. For Net-PGD, SGD (alternatively, Adam) optimizer is used for the projection step and SGD optimizer for the gradient step in Step 3 of Alg. 1 and Step 4 of Alg. 2. For implementing Lasso algorithm, Python's `sklearn.linear_model` library was used and we set the regularization factor $\alpha = 10^{-5}$. The MATLAB code for TVAL3 [25] made available on the author's website was used with its default settings.

*Running time:* We also report the average running times for different algorithms across different measurement levels for examples from MNIST is 5.86s (Net-GD), 5.46s (Net-PGD), 2.43s (Lasso-DCT), 0.82s (TVAL3). We note that the running time of both GD and PGD for CS-UNP are competitive.

## C.2    Compressive phase retrieval

*Implementation details:* For compressive phase retrieval with deep network priors, both Net-GD and Net-PGD were implemented using the PyTorch framework with Python 3 and using GPU support. All optimization procedures were implemented using SGD optimizer. For implementing Sparta algorithm, the algorithm from [22] was implemented in MATLAB.

We also report the average running times for different algorithms across different measurement levels for examples from MNIST is 25.59s (Net-GD), 28.46s (Net-PGD), 3.80s (Sparta-DCT).

*Goodness of random initialization:* Our theoretical guarantees for phase retrieval hold only as long as the initialization $x^0$ is close to the ground truth $x^*$. We perform rigorous experiments to assert that uniform random initialization of the weights $\mathbf{w}^0$ of the neural network, ensure that the initial estimate $\mathbf{x}^0 = G(\mathbf{w}^0; z)$ is good. We denote the distance of initialization as $\delta_i = \|x^0 - x^T\|/\|x^T\|$ $(x^T = \hat{x})$ and report the values of $\delta_i$ for the trials in which $\|x^T - x^*\|/\|x^*\| < 0.1$. We plot the average values of $\delta_i$ in Table 1.

Table 1: Distance of initial estimate $x^0$

| n/d | d | channel configuration | nMSE of $\hat{x}$ | average $\delta_i$ values |
|---|---|---|---|---|
| 0.2 | 784 (MNIST) | 15, 15, 10 | 0.098 | 0.914 |
| 0.5 | 784 (MNIST) | 15, 15, 10 | 0.018 | 0.942 |
| 0.4 | 12288 (CelebA) | 120, 15, 15,10 | 0.020 | 0.913 |
| 0.6 | 12288 (CelebA) | 120, 15, 15,10 | 0.015 | 0.915 |

From our observation, uniform random initialization suffices to ensure that the conditions for Theorem 2 are met and $\delta_i < 1$.

## D    Proofs and supporting lemmas

In this section we proofs for the theorems discussed in the main body of this paper as well as present supporting Lemmas.

We first discuss the set-restricted restricted isometry property.

The $(\mathcal{S}, \gamma, \beta)$ RIP holds for Gaussian matrix $A$ with high probability, as long as certain dimensionality requirements are met. We show this via Lemma 1 as follows:

**Lemma 1.** *If an image $x \in \mathbb{R}^d$ has a decoder prior (captured in set $\mathcal{S}$), where the decoder consists of weights $\mathbf{w}$ and piece-wise linear activation (ReLU), a random Gaussian matrix $A \in \mathbb{R}^{n \times d}$ with elements from $\mathcal{N}(0, 1/n)$, satisfies $(\mathcal{S}, 1 - \alpha, 1 + \alpha)$-RIP, with probability $1 - e^{-c\alpha^2 n}$, as long as $n = O\left(\frac{k_1}{\alpha^2} \sum_{l=2}^{L} k_l \log d\right)$, for small constant $c$ and $0 < \alpha < 1$.*

*Proof.* We first describe the two layer setup.

Consider the action of measurement matrix $A$ defined on vector $h$, where $h := U_1 \sigma(ZW_1)W_2$ below:

$$u = Ah = AU_1 \sigma(Z_1 W_1)W_2.$$

where $W_1^{k_1 \times k_2}$, $W_1^{k_2 \times 1}$ and $U_1^{d \times d_1}$ with $d > d_1$.

We would like to estimate the dimensionality of $A$, required to ensure that the action of $A$ on set restricted vector $h \in \mathcal{S}$, is bounded as:

$$\gamma \|h\|^2 \leq \|Ah\|^2 \leq \beta \|h\|^2$$

with high probability. To establish this, consider the following argument which is similar to the union of subspaces argument from [6].

The action of ReLU on input $(Z_1 W_1)$ partitions the input space of variable $W_1$ into a union of linear subspaces. In particular, consider a single column of $w_{1,j}$ of $W_1$, indexed by $j$, which is $k_1$ dimensional. Then, $\sigma(Z_1 w_{1,j})$ partitions the $k_1$-dimensional input space into $(d_1^{k_1})$ $k_1$-spaces. Since there are $k_2$ such columns, effectively the $k_1 \times k_2$ dimensional space of $W_1$ is partitioned into $(d_1^{k_1})^k$, $(k_1 \times k_2)$-spaces.

Then, we can consider the union of $d_1^{k_1 k_2}$ subspaces with linearized mappings of the form:

$$u_1 = A U_1 (Z_1 W_1') W_2$$

where $W_1'$ belongs to one of the $d_1^{k_1 k_2}$ subspaces and $u_1$ is the mapping corresponding to that.

If the dimensionalities are chosen such that they satisfy $d > k_2$, and $A, U_1, Z_1$ are known matrix operators, then the effectively $w^{k_1 \times 1} := W_1' W_2$ represents the accumulated action of the weights, belonging to one of the $d_1^{k_1 k_2}$ subspaces, $(U_1 Z_1)^{d \times k_1}$ is a linear transformation from a lower dimensional space to a higher dimensional space. Then, if $A$ is designed as an oblivious subspace embedding (OSE) (Lemma 3 in Appendix D) of $U_1 Z_1 w$, for a single $k_1$-dimensional subspace of $w$, one requires $m = O\left(\frac{k_1}{\alpha^2}\right)$ samples to embed the vector $w$, as

$$(1 - \alpha)\|h\|^2 \leq \|Ah\|^2 \leq (1 + \alpha)\|h\|^2, \tag{7}$$

with probability $1 - e^{-c\alpha_1^2 n}$, for constant $\alpha_1 < \alpha$. Since there are $d_1^{k_1 k_2}$ such subspaces, then for the OSE to hold for all subspaces, one requires to take a union bound as $1 - d_i^{k_i k} e^{-c\alpha_1^2 n}$. Therefore the expression in Eq. 7 holds for all $h \in \mathcal{S}$, with probability $1 - e^{-c\alpha_2^2 n}$ and $\alpha_2 < \alpha_1$. Therefore, one requires $n = O\left(\frac{k_1 k_2 \log d_1}{\alpha_1^2}\right)$, to ensure that $A$ satisfies $(\mathcal{S}, 1 - \alpha, 1 + \alpha)$-RIP with probability $1 - e^{-c\alpha_2^2 n}$.

***Multiple layers:*** A similar argument can be extended for multiple layers. Consider an $L$ layer formulation:

$$u = A U_{L-1} \sigma(\ldots \sigma(U_1 \sigma(Z_1 W_1) W_2) W_3 \ldots) W_L$$

with $W_L^{k_L \times 1}$ and $U_{L-1}^{d \times d_{L-1}}$.

The first non-linearity partitions the space into $d_1^{k_1 k_2}$ $k_1 \times k_2$-dimensional spaces. Thus we have the part-linearized mapping of the form:

$$u_1 = A U_{L-1} \sigma(\cdots U_2 \sigma(U_1 Z_1 W_1' W_2) W_3 \cdots) W_L$$

and there are $d_1^{k_1 k_2}$ of these.

The second non-linearity acts on input $(U_1 Z_1)^{d_2 \times k_1} \cdot (W_1' W_2)^{k_1 \times k_3}$ of each of these partitions, and creates more partitions; $d_2^{k_1 k_3}$ partitions of the $k_1 \times k_3$ space. This creates effectively $d_1^{k_1 k_2} \times d_2^{k_1 k_3} \leq d_2^{k_1(k_2 + k_3)}$ (since $d_2 > d_1$) partitions in total and these constitute linearized embeddings of the form:

$$u = A U_{L-1} \sigma(\ldots \sigma(U_2 U_1 Z_1 W_1' W_2') W_3 \ldots) W_L$$

where $W_1' W_2'$ belong to one of the $d_1^{k_1 k_2} \cdot d_2^{k_1 k_3}$ subspaces.

Extending the same argument to all subsequent non-linearities (total $(L - 1)$ such) and linearizing, we have mappings of the form

$$u_{L-1} = A U_{L-1}(\ldots (U_2 U_1 Z_1 W_1' W_2') W_3' \ldots) W_L$$

$$h_{L-1} = \left(\left(\prod_{l=1}^{L-1} U_l\right) Z_1\right) \cdot \left(\prod_{l=1}^{L} W_l\right)$$

$$= B \cdot w \tag{8}$$

where $B := \left(\prod_{l=1}^{L-1} U_l\right) Z_1$ and $w := \left(\prod_{l=1}^{L} W_l\right) \in \mathbb{R}^{k_1}$. The total number of partitions are $d_1^{k_1 k_2} \times d_2^{k_1 k_3} \dots d_{L-1}^{k_1 k_L} \le d^{k_1 \sum_{l=2}^{L} k_l}$, since $d > d_{L-1} > \dots d_1$, via upsampling operations. Effectively we consider a union of $d^{k_1 \sum_{l=2}^{L} k_l}$ subspaces of dimension $k_1$.

Repeating the argument from the analysis for two layers, if $A$ is designed as an oblivious subspace embedding (OSE) (Lemma 3 in Appendix D) of $B \cdot w$, for a single $k_1$-dimensional subspace of $Bw$, one requires $m = O\left(\frac{k_1}{\alpha^2}\right)$ samples to embed the vector $w$, with the bound in Eq. 7 with probability $1 - e^{-c\alpha_1^2 n}$, for constant $\alpha_1 < \alpha$.

Therefore, the embedding from Eq. 7 holds for

$$h = U_{L-1}\sigma(\dots\sigma(U_1\sigma(ZW_1)W_2)W_3\dots)W_L,$$

as long as $n = O\left(\frac{k_1 \sum_{l=2}^{L} k_l \log d}{\alpha_1^2}\right)$ , with probability $1 - e^{-c\alpha_o^2 n}$, which implies that $A$ satisfies $(\mathcal{S}, 1 - \alpha, 1 + \alpha)$-RIP with high probability.

$\square$

Next, we present some corollaries which will be useful for proving some of our theoretical claims.

**Corollary 1.** *For parameter $\alpha > 0$, if a matrix $A \in \mathbb{R}^{n \times d}$ satisfies $(\mathcal{S}, 1 - \alpha, 1 + \alpha)$-RIP with probability $1 - e^{-c\alpha_o^2 n}$, for all $x \in \mathcal{S}$, then for $x_1, x_2 \in \mathcal{S}$,*

$$(1 - \alpha)\|x_1 - x_2\|^2 \le \|A(x_1 - x_2)\|^2 \le (1 + \alpha)\|x_1 - x_2\|^2,$$

*holds with probability $1 - e^{-c_2\alpha_o^2 n}$, where $c_2 < c$.*

*Proof.* Since $x_1, x_2 \in \mathcal{S}$, both $x_1, x_2$ lie in the union of $k_1$-dimensional subspaces, the difference vector $x_3 = x_1 - x_2 \in \mathcal{S}'$, lies in a union of $2k_1$-dimensional subspaces. For $(\mathcal{S}, 1 - \alpha, 1 + \alpha)$-RIP to hold for the difference set, one continues to require $n = O\left(\frac{k_1 \sum_{l=2}^{L} k_l \log d}{\alpha_1^2}\right)$. $\square$

**Corollary 2.** *If $A$ satisfies set-restricted RIP and $h^t = x^t - x^*$, with $x^t, x^* \in \mathcal{S}$ then*

$$\|(1 - \eta A^\top A)h^t\| \le \max\{1 - \eta\lambda_{min}, \eta\lambda_{max} - 1\}\|h^t\|$$

*with $\lambda_{min} = (1 - \alpha)$ and $\lambda_{max} = (1 + \alpha)$.*

*Proof.* Consider $h \in \mathcal{S}'$, where $h = h^t = x^t - x_2$ and $x^t, x^* \in \mathcal{S}$. Then from Set-RIP and Corollary 1,

$$(1 - \alpha)\|h\|^2 \le \|Ah\|^2 \le (1 + \alpha)\|h\|^2.$$

From Eq. 8, if $x_1, x_2 \in \mathcal{S}$, then it is possible to write $h$ to arise from a union of $2k_1$-dimensional subspaces of the form $h = Bw$. Then,

$$(1 - \alpha)\|Bw\|^2 \le \|ABw\|^2 \le (1 + \alpha)\|Bw\|^2. \tag{9}$$

where $w \in \mathbb{R}^{2k_1}$. We need to evaluate the eigenvalues of $\|A^\top A\|$ restricted on set $\mathcal{S}'$, which we can do by inducing a projection on the union of subspaces $B$ as

$$\|A^\top Ah\| = \|B^\top A^\top ABw\|$$

Therefore, the minimum and maximum eigenvalues of $\|A^\top A\|$ restricted on set $\mathcal{S}'$ are

$$\sigma_{min}(AB) \le \|B^\top A^\top A\bar{B}\|_2 \le \sigma_{max}(AB)$$

Then, using Eq.9, $(1 - \alpha)\sigma_{min}(B) \le \|B^\top A^\top AB\|_2 \le (1 + \alpha)\sigma_{max}(B)$.

Since $B$ predominantly consists of a product of upsampling matrices and latent code $Z_1$, which can be always chosen such that $\sigma_{max}(Z_1) \approx \sigma_{min}(Z_1)$, therefore $\sigma_{max}(B) \approx \sigma_{min}(B) \approx 1$. $\square$

Next, we discuss the convergence of Net-PGD for compressed sensing recovery via Theorem 1.

**Theorem 1.** *Suppose the sampling matrix $A^{n \times d}$ satisfies $(\mathcal{S}, 1-\alpha, 1+\alpha)$-RIP with high probability then, Algorithm 1, with $\eta$ small enough, produces $\hat{x}$ such that $\|\hat{x} - x^*\| \le \epsilon$ and requires $T \propto \log \frac{1}{\epsilon}$ iterations.*

*Proof.* Using the definition of loss as $L(x^t) = \|y - Ax^t\|^2$,

$$
\begin{aligned}
L(x^{t+1}) - L(x^t) &= (\|Ax^{t+1}\|^2 - \|Ax^t\|^2) - 2(y^\top Ax^{t+1} - y^\top Ax^t) \\
&= \|Ax^{t+1} - Ax^t\|^2 - 2(Ax^t)^\top(Ax^t) + 2(Ax^t)^\top(Ax^{t+1}) \\
&\qquad\qquad\qquad\qquad\qquad\qquad - 2(y^\top Ax^{t+1} - y^\top Ax^t) \\
&= \|Ax^{t+1} - Ax^t\|^2 - 2(y - Ax^t)^\top(Ax^{t+1} - Ax^t) \qquad (10)
\end{aligned}
$$

We want to establish a contraction of the form $L(x^{t+1}) < \nu L(x^t)$, with $\nu < 1$.

Step 3 of Alg. 1 is solved via gradient descent:

$$
v^t = x^t - \eta A^\top (Ax^t - Ax^*) \qquad (11)
$$

Subsequently, Step 4 of Algorithm 1 learns weights $\mathbf{w}^t$ that produce $x^t = G(\mathbf{w}^t; z)$, which lies in the range of the decoder $G(\cdot)$ and is closest to the estimate $v^t$.

Step 4 of Algorithm 1 produces an update of $\mathbf{w}^t$ satisfying:

$$
\|G(\mathbf{w}^t; z) - v^t\| \le \|G(\mathbf{w}^*; z) - v^t\|
$$

Denoting $G(\mathbf{w}^t; z) := x^t$ and $G(\mathbf{w}^*; z) := x^*$, and using the update rule in Eq. 11,

$$
\begin{aligned}
\|x^{t+1} - v^t\|^2 &\le \|x^* - v^t\|^2 \\
\|x^{t+1} - x^t + \eta A^\top A(x^t - x^*)\|^2 &\le \|x^* - x^t + \eta A^\top A(x^t - x^*)\|^2 \\
\|x^{t+1} - x^t\|^2 + 2\eta(A(x^t - x^*))^\top A(x^{t+1} - x^*) &\le \|x^t - x^*\|^2 - 2\eta\|A(x^t - x^*)\|^2 \\
\tfrac{1}{\eta}\|x^{t+1} - x^t\|^2 + 2(A(x^t - x^*))^\top A(x^{t+1} - x^*) &\le \tfrac{1}{\eta}\|x^t - x^*\|^2 - 2L(x^t) \\
\implies L(x^{t+1}) + L(x^t) &\le \tfrac{1}{\eta}\|x^t - x^*\|^2 - \tfrac{1}{\eta}\|x^{t+1} - x^t\|^2 \\
&\qquad\qquad\qquad + \|A(x^{t+1} - x^t)\|^2
\end{aligned}
$$

where we have used the expansion in Eq. 10. We now use $(\mathcal{S}, \gamma, \beta)$-RIP. If a Gaussian measurement matrix is considered then $\gamma = 1 - \alpha$ and $\beta = 1 + \alpha$.

Using $(S, \gamma)$-RIP on the first term on the right side,

$$
\|x^* - x^t\|^2 \le \frac{1}{\gamma}\|A(x^* - x^t)\|^2
$$

Second, using $(S, \beta)$-RIP on the last term on the right side,

$$
\|A(x^{t+1} - x^t)\|^2 \le \beta\|x^{t+1} - x^t\|^2
$$

Accumulating these expressions and substituting,

$$
\begin{aligned}
L(x^{t+1}) + L(x^t) &\le \frac{1}{\eta\gamma}L(x^t) + \left(\beta - \frac{1}{\eta}\right)\|x^{t+1} - x^t\|^2 \\
&\overset{\beta\eta<1}{\le} \frac{1}{\eta\gamma^2}L(x^t) \\
\implies L(x^{t+1}) &\le \nu L(x^t) \\
\implies L(x^T) &\le \nu^T L(x^0)
\end{aligned}
$$

where $0 < \nu < 1$ and $\nu = \left(\frac{1}{\eta\gamma^2} - 1\right)$ and picking $\eta < 1/\beta$. Invoking $(S, \gamma, \beta)$-RIP again,

$$
\|x^T - x^*\|^2 \le \frac{1}{\gamma}\|y - Ax^T\|^2 \le \frac{\nu^T}{\gamma}\|y - Ax^0\|^2 := \epsilon
$$

Hence to reach $\epsilon$- accuracy in reconstruction, one requires $T$ iterations where

$$T = \log_\alpha \left( \frac{\|y - Ax^0\|^2}{\gamma\epsilon} \right).$$

Note that the contraction $L(x^{t+1}) \le \nu L(x^t)$ coupled with $(\mathcal{S}, \gamma, \beta)$-RIP implies a distance contraction $\|x^{t+1} - x^*\| \le \nu_o \|x^t - x^*\|$, with $\nu_o = \nu\sqrt{\beta/\gamma}$.

Step 4 of Algorithm 1, which is essentially the case of fitting a noisy image to a deep neural network prior can be solved via gradient descent. We discuss this projection in further detail in Section A. $\square$

Next, we discuss the main convergence result of Net-PGD for compressive phase retrieval in Theorem 2.

**Theorem 2.** *Suppose the sampling matrix $A^{n \times d}$ with Gaussian entries satisfies $(\mathcal{S}, 1-\alpha, 1+\alpha)$-RIP with high probability, Algorithm 2 solves Eq. 5 with $\eta$ small enough, such that $\|\hat{x} - x^*\| \le \epsilon$, as long as the weights are initialized appropriately and the number of measurements is $n = O\left( k_1 \sum_{l=2}^{L} k_l \log d \right)$.*

*Proof.* Step 4 of Algorithm 2 is solved via a variant of gradient descent called Wirtinger flow [36], which produces updates of the form:

$$\begin{aligned}
v^{t+1} &= x^t - \eta A^\top \left( Ax^t - Ax^* \circ \text{sign}(Ax^*) \circ \text{sign}(Ax^t) \right) \\
&= x^t - \eta A^\top \left( Ax^t - Ax^* \right) - \eta A^\top Ax^* \circ (1 - \text{sign}(Ax^*) \circ \text{sign}(Ax^t)) \\
&= x^t - \eta A^\top \left( Ax^t - Ax^* \right) - \eta \varepsilon_p^t
\end{aligned} \qquad (12)$$

where $\varepsilon_p^t := A^\top Ax^* \circ (1 - \text{sign}(Ax^*) \circ \text{sign}(Ax^t))$ is *phase estimation* error.

If $\text{sign}(Ax^*) \approx \text{sign}(Ax^t)$, then the above resembles the gradient step from the linear compressed sensing formulation. Thus, if $x^0$ is initialized well, the error due to phase mis-match $\varepsilon_p^t$ can be bounded, and subsequently, a convergence result can be formulated.

Next, Step 4 of Algorithm 2 learns weights $\mathbf{w}^t$ that produce $x^t = G(\mathbf{w}^t; z)$, which lies in the range of the decoder $G(\cdot)$ and is closest to the estimate $v^t$. We discuss this projection in further detail in Appendix A.

Step 4 of Algorithm 2 produces an update of $\mathbf{w}^t$ satisfying:

$$\begin{aligned}
\|G(\mathbf{w}^t; z) - v^t\| &\le \|G(\mathbf{w}^*; z) - v^t\| \\
\equiv \|x^t - v^t\| &\le \|x^* - v^t\|
\end{aligned}$$

for $t = \{1, 2, \dots T\}$. Then, the above projection rule yields:

$$\|x^{t+1} - v^{t+1} + v^{t+1} - x^*\| \le \|x^{t+1} - v^{t+1}\| + \|x^* - v^{t+1}\| \le 2\|x^* - v^{t+1}\|$$

Using the update rule from Eq. 12 and plugging in for $v^{t+1}$:

$$\frac{1}{2}\|x^{t+1} - x^*\|^2 \le \|(x^t - x^*) - (\eta A^\top \left( Ax^t - Ax^* \right) + \eta \varepsilon_p^t)\|^2$$

Defining $h^{t+1} = x^{t+1} - x^*$ and $h^t = x^t - x^*$, the above expression is

$$\frac{1}{2}\|h^{t+1}\| \le \|h^t - \eta A^\top Ah^t - \eta\varepsilon_p^t\| \le \|(1 - \eta A^\top A)h^t\| + \eta\|\varepsilon_p^t\| \qquad (13)$$

We now bound the two terms in the expression above separately as follows. The first term is bounded using matrix norm inequalities Using Corollary 2 (in Appendix D) of $(\mathcal{S}, \gamma, \beta)$-RIP:

$$\|(1 - \eta A^\top A)h^t\| \le \max\{1 - \eta\lambda_{min}, \eta\lambda_{max} - 1\}\|h^t\|$$

where $\lambda_{min}$ and $\lambda_{max}$ are the minimum and maximum eigenvalues of $A^\top A$ restricted on set $\mathcal{S}$, and via Corollary 2, $\lambda_{min} = (1 - \alpha)$, $\lambda_{max} = (1 + \alpha)$.

Hence the first term in the right side of Eq.13 is bounded as:

$$\|(1 - \eta A^\top A)h^t\| \le \rho_o\|h^t\|.$$

where $\rho_o = \max\{1 - \eta(1 - \alpha), \eta(1 + \alpha) - 1\}$. The second term in Eq.13 is bounded via Lemma 2 as follows:

$$\|\varepsilon_p^t\| \leq \delta_o \|x^t - x^*\|$$

as long as $\|x^0 - x^*\| \leq \delta_i \|x^*\|$.

Substituting back in Eq.13,

$$\|x^{t+1} - x^*\| \leq 2(\rho_o + \eta\delta_o)\|x^t - x^*\| := \rho\|x^t - x^*\|.$$

Then, if we pick constant $\eta = \frac{1}{1+\alpha+1-\alpha} = 1$ that minimizes $\rho := 2(\max\{1 - \eta(1 - \alpha), \eta(1 + \alpha) - 1\} + \eta\delta_o)$, to yield $\rho = 2(\alpha + \delta_o)$ then we obtain the linear convergence criterion as follows:

$$\|x^{t+1} - x\| \leq \rho\|x^t - x\|.$$

Here, if we set $\alpha = 0.1$ and $\delta_o = 0.36$ from Lemma 2, then $\rho = 0.92 < 1$. Note that this proof relies on a bound on the phase error $\|\varepsilon_p^t\|$ which is established via Lemma 2 as follows:

**Lemma 2.** *Given initialization condition $\|x^0 - x^*\| \leq \delta_i \|x^*\|$, then if one has Gaussian measurements $A \in \mathbb{R}^{n \times d}$ such that $n = O\left(k_1 \sum_{l=2}^{L} k_l \log d\right)$, then with probability $1 - e^{-c_2 n}$, the following holds:*

$$\|\varepsilon_p^t\| = \|A^\top A x^* \circ (1 - sign(Ax^*) \circ sign(Ax^t))\| \leq \delta_o \|x^t - x^*\|$$

*for constant $c_2$ and $\delta_o = 0.36$.*

*Proof.* We adapt the proof of Lemma C.1. of [30] as follows.

We define indicator function $\mathbf{1}_{(a_i^\top x^t)(a_i^\top x^*)<1} = \frac{1}{2}(1 - sign(Ax^*) \circ sign(Ax^t))$ with zeros where the condition is false and ones where the condition is true.

Then we are required to bound the following expression:

$$\|\varepsilon_p^t\|^2 = 2\sum_{i=1}^{n}(a_i^\top x^*)^2 \cdot \mathbf{1}_{(a_i^\top x^t)(a_i^\top x^*)<1} \leq \delta_o^2 \|x^t - x^*\|^2$$

Following the sequence of arguments in Lemma C.1. of [30] (or Lemma C.1 of [37]), one can show that for a *given* $x^t$,

$$\|\varepsilon_p^t\|^2 \leq \delta_o^2 + \kappa + \frac{3c_1\kappa}{\delta_i} < 0.13 + \kappa + \frac{3c_1\kappa}{\delta} \tag{14}$$

with high probability, $1 - e^{-cn\kappa^2}$, for small constants $c, c_1, \delta$, as long as $\|x^t - x^*\| \leq 0.1\|x^*\|_2$. Here the bound on $\delta_o^2$ (in this case 0.13) is a monotonically increasing function of the distance $\delta_i^t = \frac{\|x^t - x^*\|_2}{\|x^*\|_2}$.

If the projected gradient scheme produces iterates satisfying

$$\|x^{t+1} - x^*\| < \rho\|x^t - x^*\|$$

with $\rho < 1$, then the condition in Eq. 14 is satisfied for all $t = \{1, 2, \ldots T\}$ as long as the initialization $x^0$ satisfies $\|x^0 - x^*\| \leq 0.1\|x^*\|_2$ (i.e. $\delta_i^0 := \delta_i = 0.1$).

Now, the expression in Eq. 14 holds for a fixed $x^t$. To ensure that it holds for all possible $x \in \mathcal{S}$, we need to use an epsilon-net argument over the space of variables spanned by $\mathcal{S}$. The cardinality of $\mathcal{S}$ is

$$\text{card}(\mathcal{S}) < d^{k_1 \sum_{l=2}^{L} k_l}$$

as seen from the derivation of RIP in Lemma 1. Therefore,

$$\|\varepsilon_p^t\| \leq 0.13 + \kappa + \frac{3c_1\kappa}{\delta_i}$$

with probability $1 - d^{k_1 \sum_{l=2}^{L} k_l} e^{-cn\kappa^2}$ for small constant $c$. To ensure that high probability result holds for *all* $x \in \mathcal{S}$,

$$e^{k_1 \sum_{l=2}^{L} k_l \log d} e^{-cn\kappa^2} < e^{-c_2 n}$$

$$k_1 \sum_{l=2}^{L} k_l \log d - cn\kappa^2 < -c_2 n$$

$$n > \frac{1}{c\kappa^2 - c_2} k_1 \sum_{l=2}^{L} k_l \log d > c_3 k_1 \sum_{l=2}^{L} k_l \log d$$

for appropriately chosen constants $c, c_2, c_3$. $\qquad\square$

Note that this Theorem requires that the weights are initialized appropriately, satisfying $\|x^0 - x^*\| \leq \delta_i \|x^*\|$. In Section 6 we perform rigorous experiments to show that random initialization suffices to ensure that $\delta_i$ is small. $\qquad\square$

Finally we state the statement for Oblivious Subspace Embedding, which is the core theoretical lemma required for proving our RIP result.

**Lemma 3.** *Oblivious subspace embedding (OSE) [38]. A $(k, \alpha, \delta)$-OSE is a random matrix $\Pi^{n \times d}$ such that for any fixed $k$-dimensional subspace $\mathcal{S}$ and $x^{d \times 1} \in \mathcal{S}$, with probability $1 - \delta$, $\Pi$ is a subspace embedding for $S$ with distortion $\alpha$, where $n = O(\alpha^{-2}(k + \log(\frac{1}{\delta})))$.*

*The failure probability is $\delta = e^{-cn\alpha^2 + ck}$, for small constant $c$ and the embedding satisfies:*

$$(1 - \alpha)\|x\|^2 \leq \|\Pi x\|^2 \leq (1 + \alpha)\|x\|^2.$$