[Reviews · NeurIPS 2019]

Reviewer 1



This paper presents compressed sensing style recovery guarantees for solving inverse imaging problems with untrained network priors, as initially investigated (empirically) in [9] and [10]. The work extends RIP-like conditions for exact recovery under a “deep decoder” prior for the image [1] (i.e., assumes the image is in the range of a convolutional neural network of a certain structure). In particular, they prove for a d-dimensional image, n x d Gaussian random matrices satisfy an RIP condition for images in the range of deep decoder network provided n is on the order of the dimension of the initial layer of the network. Using this RIP property they prove that a projected gradient descent algorithm converges linearly to a solution of the optimization problem with the deep decoder prior. The paper also extends this analysis to an algorithm for solving compressive phase retrieval. This paper makes several excellent contributions. To my knowledge, this is the first paper to give compressed sensing style recovery guarantees for solving inverse problems with deep network priors like those in [9] and [10]. Also, to my knowledge, this is the first work applying these methods to the compressive phase retrieval problem. The paper is very clearly written. In particular, the proof sketches give useful details for understanding how the main results were obtained, and indicate clearly what earlier results they build off of (namely, ref [6]). Generally my appraisal is very positive, and I think this paper is a clear accept. However, I see some minor limitations of this work. One limitation with the present analysis (acknowledged by the authors) is that the algorithm studied is somewhat idealized in that it assumes the projection onto the range of the deep net can be computed exactly. This problem is non-convex, and standard solvers for this subproblem, such as gradient descent, could get stuck in local minima. It would be interesting if the present analysis could be extended to the case where this subproblem is only solved inexactly, perhaps even with one gradient step. Also, the sampling complexity n = O(k_1(k_2 + k_3 + ... + k_L)log d) seems suboptimal -- is the dependence on the size of intermediate layers k_2, k_3,...k_L truly necessary or an artifact of the proof technique? Some comments on this might be helpful for the reader to understand if the results are order-wise optimal or not.

Reviewer 2



Authors could further explain why should the Assumption in Def 1 hold for natural images e.g. for the considered architecture in eq3? Also, If the model approximates natural images e.g. similar to wavelets, what's the approximation bound based on DIP size {k_l, d_l}? Why DIP approach could beat compressed sensing algorithms using learned priors? Theorem1 and its proof are not very new. Results for iterative hard thresholding for sparse coding and its generalised variants for nonconvex signal constraints, cover similar claims in Theorem 1. Also theorem 1 is said to guarantee image recovery by algorithm 1. But algorithm 1 itself has a line 4 which we do not know how to exactly solve it i.e. the projection step (similar for theorem 2). Therefore global convergence guarantees are not really meaningful/practical to be discussed. What is the advantage of NetPGD over NetGD? experiments show similar runtime although NetPGD may require solving subproblems at each iteration. Further experiment details could make this point clear. Statement of Theorems 1&2: there should be a condition on step size somewhere Line 462 (typo) two W1s

Reviewer 3



This submission aims to establish recovery guarantees for deep image prior methods used in solving (linear) inverse problems. Motivated by compressive imaging and phase retrieval, a projection gradient descent (PGD) approach is adopted that entails projecting onto the range of deep networks. For Gaussian measurement matrices the convergence of the PGD method to the ground truth is established for compressive imaging as well as the phase retrieval observation models. Strong points: -recovery guarantees for deep image prior is a very important problem -the technical analysis is solid Weak points: -the novelty of the proof technique seems to be incremental; it seems to be a combination of the results in [Bora et al’17] for RIP and [Oymak et al’17] (not cited, listed below) for linear convergence of linear inverse problems with nonconvex regularization. The authors need to clarify the challenges that deep image prior model introduces. [Oymak et al’17] Oymak S, Recht B, Soltanolkotabi M. Sharp time–data tradeoffs for linear inverse problems. IEEE Transactions on Information Theory. 2017 Nov 14;64(6):4129-58. -the experiments are not convincing; they do not support the claims reported in this submission; the performance of the deep image prior is already reported in terms of improved reconstruction quality compared with the existing untrained schemes; it would be of interest to empirically assess the linear convergence behavior -the satisfaction of RIP conditions is also very important to verify; given the dataset of images sampled from the prior and the computational platforms with deep learning APIs it would be very useful to evaluate approximations of RIPs for the considered examples

[Author Response · NeurIPS 2019]

We thank the reviewers for their insightful comments.

**Rev 1**: *Projection step*: Thanks for reading the paper carefully, and for your kind assessment. We agree with you that
linear convergence of Net-PGD only holds as long as the projection step can be solved exactly (proof lines [517,543]).
Performing such a projection can be challenging in general, and resolving this is an important open problem. However,
we make two comments. First, our theory goes through even if we relax the projection requirement to be approximate,
as long as the approximation error is additively bounded, i.e. $\|x - v^t\| \leq \|x - v^*\| + \epsilon$ for some global parameter
$\epsilon > 0$. Second, there could be other relaxations; a recent (June 2019) preprint by Gamez, Eftekhari, and Cevher gives a
polynomial-time ADMM-type algorithm for inverse imaging with (trained) generative priors, as long as the mapping
$z \rightarrow x$ is near-isometric. A similar result may be possible in our (untrained) setting, but its proof will require some care.

*Tightness of sample complexity bounds*:We have not attempted to derive Fano-style information-theoretic lower bounds,
but intuitively, our sample complexity result is not too loose. If we assume $k_1=k_2\ldots=k_L$(as in [10]), then our derived
sample complexity matches $N_w=\sum_{i=1}^{L-1} k_i k_{i+1}$(no. of unknown parameters of network prior) up to log factors. Our
result is asymmetric in $k_1$ which makes sense as $k_1$ is dim. of latent code, but this could be an artifact of proof technique.

**Rev 2:** *Novelty of proofs*: We agree that IHT-style algorithms and proofs are not new; however, we emphasize that the
proof of Lemma 1 (RIP of Gaussian matrices) is novel for deep *untrained* network priors. Moreover, compressive
phase retrieval is a nonlinear forward model, and to show linear convergence we require Lemma 2, which is also novel.
Theorems 1 and 2 strictly use Lemmas 1 and 2 to complete the algorithmic guarantees.

*Validity of Definition 1 and approximation error*: While the true validity of any model can be questioned, we point to
prior work in ([10] and [9]) which establish this architecture as a useful prior for natural images. Empirically, we show in
Column 2 of Figs. 1a,1b,2a,2b, that all test images are well-reconstructed using this prior. If $x^* = G(\mathbf{w}^*; z) + \varepsilon_o$ where
$G$ is a DIP parameterized by $\{k_l, d_l\}$, then the modeling error $\varepsilon_o$ gets reflected in the final reconstruction accuracy:
$\|\widehat{x} - x^*\| \leq (1 + 2\beta/\gamma) \|\varepsilon_o\| + \varepsilon/\gamma$ whenever $\|y - A\widehat{x}\| \leq \min_{x \in \mathcal{S}} \|y - Ax\| + \varepsilon$ (follows from combination of
Lemma 1 and Lemma 4.3 of [6]), as long as sample requirements for Lemma 1 are satisfied. We will append this result.

*Improvements over learned image priors and AMP*: Since our network prior is untrained, we do not claim accuracy
benefits over learning-based methods; the (significant) benefit of our approach is that it does not require large training
datasets. We will certainly add additional comparisons to AMP; however, BM3D-AMP appears to perform worse than
TVAL3 (Figs. 1,2 of concurrent work in [17]) in extremely low sample regimes such as those in this paper.

*Advantage of Net-PGD v/s Net-GD:* We do not know how to analyze NetGD, since the output of the intermediate
iterations is not guaranteed to lie in the range of untrained generators (which our theoretical analysis requires). In
our experience, the running times of Net-GD and Net-PGD are comparable; even though Net-PGD requires solving
subproblems in each iteration, the overall iteration complexity is lower. We will clarify this in the revision. Step size
requirements will be appended to Theorems 1 and 2, as indicated on Lines [524] and [552].

**Rev 3:** *Novelty over Bora et al, '17, Oymak et al '17*: We respectfully push back against novelty criticisms in general,
and specifically when contrasted against these two papers. Please allow us to clarify possible misunderstandings.

First, we emphasize that our second application (compressive phase retrieval) is a **non-linear** inverse problem. To our
knowledge, we are the first to formally consider deep image priors in nonlinear recovery problems (and phase retrieval
in particular) whereas these previous papers only address linear inverse problems.

Second, we emphasize the DIP model is **not** a generative prior model a la Bora et al '17. They assume a trained
network (and optimize over the latent code), while we assume an untrained network with a fixed, random latent code
and optimize over all *network weights*. This obviously is a much more challenging problem experimentally, but also
theoretically. Therefore, our results and techniques, used to establish Lemma 1 and particularly Lemma 2, are more
involved than those in Bora et al, '17.

Third, our motivation, techniques, and results are very different from the approach in the seminal work of Oymak et
al, '17. They focus on linear inverse problems and priors defined by *convex* constraint sets; moreover, their focus is
on getting *sharp* bounds which they succeed to do using their Gaussian widths analysis. In contrast, our proofs are
significantly simpler and shorter (albeit potentially sub-optimal; see our response to Reviewer 1 above).

*Validating the RIP result*: It is well-known that RIP is empirically difficult to verify for any given measurement matrix
(for the normal sparsity case, it is known to be NP-hard). Moreover, RIP is a sufficient but not necessary condition for
successfully solving any inverse problem. The tradition in the literature has been to experimentally measure sample
complexity and show improvement over handcrafted priors, which we have presented in Figures 1c and 2c.

*More empirical results*: We will gladly add more experiments to validate local linear convergence of
Alg.1. (see right, log scale) and Alg.2. Please also note that we show superior empirical performance
of Alg. 2 over a state-of-art (Sparta) for compressive phase retrieval (Fig.2), validating our theory.



[Meta-Review · NeurIPS 2019]

This paper describes new theory associated with using untrained neural networks to conduct regularization. The idea has been explored empirically before, but the theory provides new insight into why these ideas work. Some of the proofs are derivative of proofs in other contexts, and the key assumption (analogous to the RIP) is strong, especially since in the deep image prior makes this assumption much less interpretable than it would be in a sparse recovery problem.